# RPG360: Robust 360 Depth Estimation with Perspective Foundation Models and Graph Optimization

**Dongki Jung   Jaehoon Choi   Yonghan Lee   Dinesh Manocha**
University of Maryland, College Park
{jdk9405, kevchoi, lyhan12, dmanocha}@umd.edu

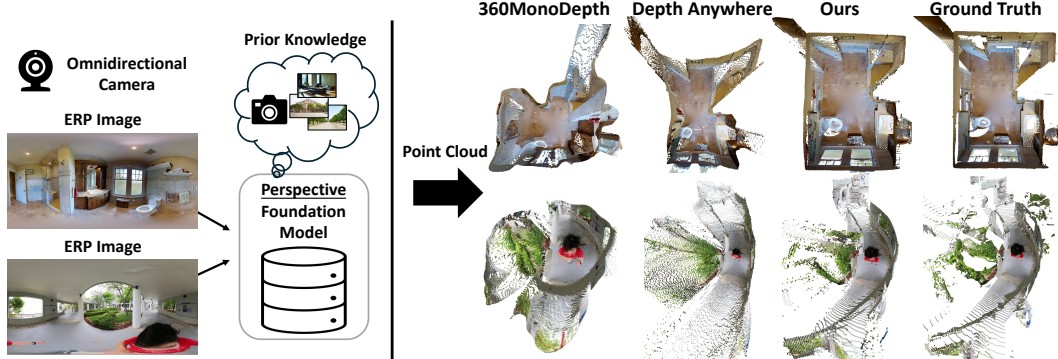

Figure 1: (Left) 360° images can be transformed into multiple undistorted tangential plane images, and then scale-ambiguous depth maps can be predicted using perspective-foundation models [46, 68, 25]. (Right) 360MonoDepth [47] and Depth Anywhere [60], have addressed the scale ambiguity issue using optimization or learning-based methods. However, these approaches suffer from 3D structural degradation, as seen in the reconstructed point cloud visualization. Our proposed method leverages the prior knowledge of the perspective foundation model along with graph optimization, thereby enhances 3D structural awareness and achieves superior performance.

## Abstract

The increasing use of 360° images across various domains has emphasized the need for robust depth estimation techniques tailored for omnidirectional images. However, obtaining large-scale labeled datasets for 360° depth estimation remains a significant challenge. In this paper, we propose RPG360, a training-free robust 360° monocular depth estimation method that leverages perspective foundation models and graph optimization. Our approach converts 360° images into six-face cubemap representations, where a perspective foundation model is employed to estimate depth and surface normals. To address depth scale inconsistencies across different faces of the cubemap, we introduce a novel depth scale alignment technique using graph-based optimization, which parameterizes the predicted depth and normal maps while incorporating an additional per-face scale parameter. This optimization ensures depth scale consistency across the six-face cubemap while preserving 3D structural integrity. Furthermore, as foundation models exhibit inherent robustness in zero-shot settings, our method achieves superior performance across diverse datasets, including Matterport3D, Stanford2D3D, and 360Loc. We also demonstrate the versatility of our depth estimation approach by validating its benefits in downstream tasks such as feature matching $3.2 \sim 5.4\%$ and Structure from Motion $0.2 \sim 9.7\%$ in AUC@5°. Project Page: https://jdk9405.github.io/RPG360/

39th Conference on Neural Information Processing Systems (NeurIPS 2025).

# 1  Introduction

360° sensors offer significant advantages due to their wide field of view, allowing the capture of rich contextual information within a single image [74, 9, 22, 66]. The growing adoption of such panoramic images in various applications, such as robot navigation [38, 65], virtual reality [35], autonomous vehicles [41], and immersive media [10], necessitates robust depth estimation, generalizing to zero-shot settings, techniques specifically tailored for 360° images. However, applying conventional perspective depth estimation methods [67, 68, 71, 25, 62] to 360° images directly is challenging due to fundamental differences in camera models, which introduce distortions. To address these challenges, previous research in 360° depth estimation has broadly followed two approaches: 1) learning-based methods and 2) optimization-based methods.

Existing *learning-based approaches* [78, 43, 73, 54, 27, 59, 2, 1, 77, 72, 28] have focused on designing 360° depth estimation networks trained on labeled 360° datasets. These methods attempt to mitigate distortions through adaptive network architectures [78, 53, 61], or by leveraging cubemap [58, 59, 27] and tangent image [37, 51] projections during training. However, a major limitation of these approaches is the scarcity of labeled datasets for 360° depth estimation. A few works have proposed self-supervised training methods [79] or generating pseudo ground truth depth maps [60] to overcome this limitation.

In contrast, *optimization-based methods leveraging perspective foundation models* [47, 42] offer an alternative approach that does not require labeled datasets. Recently, monocular depth estimation methods based on foundation models in the perspective image domain [45, 67, 68, 32, 71, 25, 62] have demonstrated robust performance in zero-shot scenarios. 360° images can be transformed into multiple tangential plane images to reduce distortions. This transformation enables respective depth and normal estimation on each tangential plane, which can then be recombined into a single equirectangular projection. Leveraging pre-trained robust foundation models helps address the limitations associated with the lack of labeled 360° datasets. However, a key challenge remains: Monocular depth estimation encounters scale ambiguity when merging independently estimated depth maps from different viewpoints. While several methods [47, 42] for 360° depth estimation propose blending overlapped regions of tangent-plane depth maps from perspective networks, this simple blending overlooks crucial 3D structural information, as shown in Fig. 1. Although the depth maps may exhibit high quality in 2D space, their corresponding 3D point clouds often suffer from 3D structural inconsistencies and performance degradation.

**Main Results:**   We present a novel structure-aware optimization technique for 360° depth estimation using perspective foundation models [71, 25, 11]. Our approach converts an equirectangular projection image into perspective images via cubemap projections, which represent the 360° scene without overlapping regions. Depth and normal estimation are performed using a perspective foundation model. Since monocular depth estimation provides only relative depth information, the depth scales across different faces of the cubemap may be inconsistent. To address this issue, we introduce a novel graph optimization approach that parameterizes the predicted depth maps and surface normals with additional per-face scale parameters. This parameterization ensures depth scale consistency across different perspective images of the cubemap. Our approach demonstrates robust depth estimation performance across diverse datasets, including indoor and outdoor environments such as Matterport3D [5], Stanford2D3D [4], and 360Loc [26]. The main contributions include:

- We propose a robust optimization-based 360° depth estimation method leveraging perspective foundation models to address the scarcity of labeled 360° dataset.

- We introduce a graph optimization formulation that integrates depth and surface normals with additional per-view scale parameters to enforce depth scale consistency. This optimization preserves the 3D structure and demonstrates superior performance in terms of 3D evaluation metrics.

- Most learning-based methods exhibit performance degradation when trained on a dataset that differs from the test scenes. In contrast, our method does not require a training dataset and is unaffected by domain gap.

- We demonstrate the versatility and benefits of our 360° depth estimation approach by applying it to downstream tasks such as feature matching [30] and structure-from-motion [31].

## 2 Related Work

### 2.1 Perspective Foundation Models

Foundation models possess remarkable adaptability, allowing them to generalize effectively across different contexts. CLIP [44] has exhibited outstanding versatility across a wide range of vision-language tasks. DINOv2 [39] introduced a powerful visual encoder designed for various downstream 2D vision applications, such as segmentation, object detection, and depth estimation. DUSt3R [63] further explores the applicability of foundation models in 3D vision tasks, leveraging CroCo [64] pre-training to enhance performance. Furthmore, learning-based monocular depth estimation methods have evolved from optimizing training techniques for various model architectures [12, 16, 19, 20, 6–8] to enhancing zero-shot capabilities [45, 46, 11, 67, 68, 32, 71, 25], enabling strong generalization across diverse domains. Notably, Metric3D [71, 25] extends this direction by specifically exploring zero-shot performance for metric depth estimation and surface normal prediction.

### 2.2 360° Depth Estimation

360° depth estimation suffers from the inherent distortions introduced by the equirectangular projection. OmniDepth [78] leverage Spherical Convolutions [53] to mitigate this projection distortion. Other methods improve performance by combining multiple projection techniques, such as fusing equirectangular and cubemap projections [58, 59, 27] or adopting icosahedral projections [1]. Additionally, dilated convolutions have been explored to enhance feature extraction in distorted panoramic images [77]. Gravity-aligned feature learning has also been proposed to improve 360° depth estimation [54, 43]. Meanwhile, transformer-based approaches [28, 73, 2, 51] have recently gained traction for capturing global panoramic context more effectively. Despite these advancements, supervised learning methods for 360° depth estimation still heavily rely on labeled datasets. Due to the lack of large-scale real-world annotations, these methods are often dependent on synthetic datasets [11, 75, 3], potentially causing performance degradation in real-world scenarios.

To mitigate this issue, some works explore self-supervised learning via view synthesis [57, 79]. Another line of research focuses on leveraging pre-trained perspective monocular depth estimation models for panorama depth estimation. For example, the pre-trained weights of a perspective monocular depth estimation model are fine-tuned on 360° images using distortion-aware convolutional filters [56]. More recently, perspective foundation models have been incorporated, alleviating the dependency on large-scale labeled 360° datasets. 360MonoDepth [47] utilizes foundation models trained on perspective images [45, 46] to predict depth on tangent images, which are later merged using a blending algorithm. Depth Anywhere [60] adopts a different approach by predicting depth for cubemap images using a foundation model [67] and generating pseudo ground truth, which is then refined through affine-invariant loss. However, while these methods incorporate certain distortion-aware techniques, they lack explicit consideration of 3D structural consistency in the estimated depth maps. We propose a novel approach that leverages perspective foundation models to estimate depth while explicitly promoting 3D structure awareness for 360° depth estimation.

## 3 Our Approach: RPG360

The overall process is illustrated in Fig. 3. In Section 3.1, the Equirectangular Projection (ERP) image captured by a 360° camera is first converted into distortion-free perspective (PER) images for individual monocular depth estimation. We adopt a cubemap projection to represent PER images due to its computational efficiency, as every pixel is treated as a parameter and the cubemap representation avoids overlapping regions. The estimated depth maps are then merged back into the ERP domain using the refinement method described in Section 3.2.

### 3.1 Transformation of Coordinate Systems

An ERP image is a 2D plane representation that depicts normalized rays on a spherical surface in a flattened form. The coordinate transformation can be formulated as $\boldsymbol{\xi} = \boldsymbol{\pi}(\boldsymbol{S})$, where $\boldsymbol{\xi} = (\theta, \phi)$ denotes the spherical coordinates with $\theta \in [-\pi, \pi]$, $\phi \in [-\frac{\pi}{2}, \frac{\pi}{2}]$. The corresponding 3D Cartesian coordinates are given by the unit vector $\boldsymbol{S} = (S_x, S_y, S_z)$. For

PER images obtained from a six-face cubemap projection, $c \in \{0, ..., 5\}$ indicates the corresponding face of the cubemap. The pixel coordinates of PER images are denoted as $p_i$,

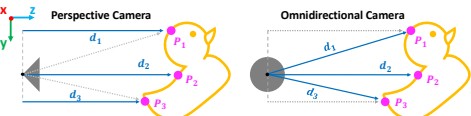

Figure 2: Differences in depth definitions between camera models. (Left) In a perspective camera, depth is defined as the distance along the z-axis from the camera center. (Right) In an omnidirectional camera, depth is measured as the Euclidean distance from the camera center to the 3D point.

$$\begin{cases} S_x = \sin(\theta)\cos(\phi) \\ S_y = \sin(\phi) \\ S_z = \cos(\theta)\cos(\phi), \end{cases} \quad \widetilde{p}_c = K \cdot R_c^T \cdot S. \quad (1)$$

where $\widetilde{p}_c$ is the homogeneous coordinate of $p_c$. The rotational extrinsic parameter for each face $i$ is represented as $R_c$, while $K \in \mathbb{R}^{3\times3}$ denotes the intrinsic camera matrix, which sets the focal length and the principal point at half of the image size. Using this coordinate transformation, we can convert the depth maps between PER and ERP space. Notably, the definition of depth differs between perspective cameras and omnidirectional cameras. As illustrated in Fig.2, depth in PER cameras is measured as the distance along the z-direction, whereas in 360° cameras, it represents the radial distance from the sensor. Thus, a scaling factor $\rho$ should be applied during the warping process to compensate for the variation in depth definitions,

$$D^{ERP}\langle\xi\rangle = \rho D_c^{PER}\langle p_c\rangle, \qquad \rho = \|K^{-1} \cdot \widetilde{p}_c\|. \quad (2)$$

where $D_i^{PER}$ is the predicted depth map from the pre-trained PER depth estimation model [25] and $\langle\cdot\rangle$ represents the nearest neighbor interpolation. We can obtain 3D points $P$ from the ERP depth map $D^{ERP}$,

$$P = D^{ERP}S. \quad (3)$$

## 3.2 Graph-based Optimization for Scale Alignment

**Scale and Shift Alignment** Monocular depth estimation inherently suffers from scale ambiguity, leading to inconsistent depth scales across different PER images. To address this inconsistency, previous studies have proposed methods such as estimating scale and shift between frames [45, 70, 23]. This can be formulated as a linear equation,

$$D^* = \lambda D + \tau. \quad (4)$$

However, ensuring consistent depth $D^*$ across different frames is challenging, as we rely on single scalar values $\lambda \in \mathbb{R}$ and $\tau \in \mathbb{R}$ for all pixels in the depth map $D$.

**Graph Optimization for Local Scale Alignment** We employ a graph optimization algorithm that parameterizes the predicted depth and normal maps. This optimization refines the depth maps by adjusting the local scale, which is analogous to the shift term, but the adjustment is applied to each pixel rather than relying on a single scalar value. We define this optimization function based on a simple equation that represents a plane defined by $(P_0, n_0)$,

$$n_0 \cdot (P - P_0) = 0, \quad (5)$$

where $P_0$ is a 3D point and $n_0$ is its corresponding normal vector of the surface. $P$ indicates any point lying on the same plane. Inspired by [50], we define the 3D points corresponding to each pixel in the predicted depth maps as nodes and assign edge weights based on the likelihood that two adjacent points belong to the same plane,

$$L_p = \sum_i \sum_{j\sim i} w_{ij}\|n_i \cdot (P_j - P_i)\|_2 + \alpha \sum_i \sum_{j\sim i} w_{ij}\|n_j - n_i\|_2, \quad (6)$$

where $j \sim i$ denotes the neighbor pixels of $i$ in the graph, and $w_{ij}$ represents the edge weight between pixel $i$ and its adjacent pixel $j$ in the ERP space. We set $\alpha$ as 0.5. In Eq. 6, the first term ensures two points are positioned in the same plane, while the second term ensures that neighboring points maintain a consistent plane based on the likelihood $w_{ij}$ between adjacent pixels. Based on the assumption that areas with the same texture correspond to the same object surface [34, 14, 49, 50], we define the edge weights using local patches and the spatial distance between pixels,

$$w_{ij} = \exp\left(-\frac{\|Q_i - Q_j\|_F^2}{2\sigma_{int}^2}\right)\exp\left(-\frac{\|i - j\|_2^2}{2\sigma_{spa}^2}\right), \quad \text{where} \quad j_x \equiv j_x \bmod W, \; j_y = j_y, \quad (7)$$

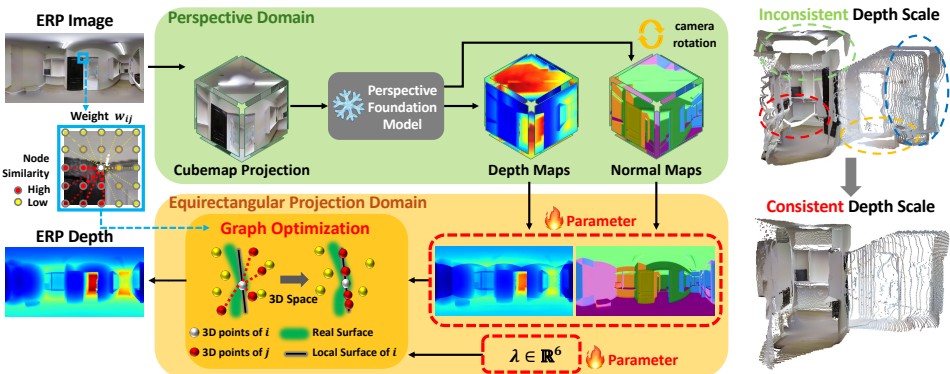

Figure 3: Overview of our depth estimation method. **1)** The input equirectangular (ERP) image is projected onto a six-face cubemap, where each non-overlapping face is independently processed by a perspective foundation model to generate depth and normal maps. The normal maps are transformed into a common coordinate system and merged into a single ERP normal map. However, due to the inherent limitations of monocular depth estimation, the merged depth maps exhibit depth scale inconsistencies across faces. **2)** To resolve this issue, the merged depth and normal maps are parameterized, incorporating an additional scale parameter $\lambda$. Using these parameters, we perform graph optimization under a local planar approximation to obtain a scale-aligned ERP depth map. For the construction of graph, we use the intensity values and the distances between a pixel $i$ (white node) and its adjacent pixels $j$ (red nodes) to define the edge weights $w_{ij}$. Based on these weights, 3D points are aligned on the surface using both depth and normal information.

where $\mathbf{Q}_i$ represents a patch centered at pixel $i$ of the image, $\|\cdot\|_F$ denotes the Frobenius norm, and $\sigma_{int}$ and $\sigma_{spa}$ are set to 0.07 and 3.0, respectively. Since an ERP image has its left and right extremities aligned, the $x$-coordinate $j_x$ of pixel $j$ should be updated using the modulo operation with the ERP image width $W$ to ensure periodic boundary conditions.

**Per-face Parameter for Global Scale Alignment**  Simply using Eq. 6 may result in over-smoothed depth propagation in neighboring points, as shown in Fig. 6. Therefore, we exploit regularization terms for depth and normal maps to preserve the high fidelity of the input geometry. Here, we incorporate the per-face scale parameter $\lambda_c \in \mathbb{R}^6$ to achieve global scale alignment for the depth of each cubemap face. This additional parameter $\lambda_c$ functions similarly to the scale term in Eq. 4. The per-face scale parameter $\lambda_c$ is expanded to match the size of each cubemap face, and then transformed and integrated into the ERP space as a scale map $\boldsymbol{\lambda} \in \mathbb{R}^{H \times W}$.

$$L_d = \sum_i |\boldsymbol{D}_i - \boldsymbol{\lambda}\overline{\boldsymbol{D}}_i|\boldsymbol{m}_i, \quad L_n = \sum_i |\boldsymbol{n}_i - \overline{\boldsymbol{n}}_i|\boldsymbol{m}_i, \tag{8}$$

where $\overline{\boldsymbol{D}}$ and $\overline{\boldsymbol{n}}$ denote input ERP depth and normal maps. The confidence masks $\boldsymbol{m}$ are determined based on the cosine similarity between the input normal and the normal computed from the input depth map using the 8-neighbor convention [69, 29]. The total loss function is formulated as a weighted sum of the proposed loss terms,

$$L_{total} = \eta_p L_p + \eta_d L_d + \eta_n L_n, \tag{9}$$

where $\eta_p, \eta_d, \eta_n$ are weights selected through grid search.

## 4 Experiments

### 4.1 Experiments Settings

**Datasets and Evaluation Metrics**  We conduct extensive experiments on benchmark datasets — **Matterport3D** [5], **Stanford2D3D** [4], and **360Loc** [26] — using images of a resolution $1024 \times 512$. For Matterport3D and Stanford2D3D, we adopt the official train and test splits. We additionally evaluate on Matterport3D-2K ($2048 \times 1024$) for high-resolution benchmarks, following [47]. 360Loc consists of four scenes, including both indoor and outdoor environments, each providing panoramic sequences of mapping and query images. We employ all mapping sequences from 360Loc, which include ground truth poses and depth maps, to evaluate the zero-shot performance of depth estimation. As monocular depth estimation increasingly emphasizes 3D

Table 1: Quantitative comparison on the Matterport3D (**M**) and Stanford2D3D (**S**) test set. **M**$^+$ indicates the combination of **M** and Structured3D [75]. We evaluate 360° depth estimation performance under three different settings: (a) when the training and test scenes are identical, (b) when the training and test scenes differ, and (c) in a training-free setup. The best methods for each category are shown in bold faces. Most learning-based methods exhibit performance degradation when trained on a dataset that differs from the test scenes. In contrast, the optimization-based methods using perspective foundation models (🤖) do not require a training dataset, making them unaffected by domain gap. Under the training-free setting, RPG360 improves the Chamfer distance by 46.7% on Matterport3D (**M**) and 64.2% on Stanford2D3D (**S**), compared to 360MonoDepth.

| | Method | Dataset Train → Test | 3D metrics Chamfer ↓ | F-Score ↑ | IoU ↑ | | Method | Dataset Train → Test | 3D metrics Chamfer ↓ | F-Score ↑ | IoU ↑ |
|---|---|---|---|---|---|---|---|---|---|---|---|
| (a) | SliceNet [43] | M → M | 1.197 | 21.247 | 13.104 | (a) | SliceNet [43] | S → S | **0.201** | **70.700** | **57.453** |
| | UniFuse [27] | M → M | 0.444 | 42.721 | 28.573 | | UniFuse [27] | S → S | 0.331 | 43.481 | 28.555 |
| | ACDNet [77] | M → M | 0.584 | 44.863 | 30.785 | | ACDNet [77] | S → S | 0.271 | 56.832 | 42.007 |
| | BiFuse++ [59] | M → M | 0.834 | 39.383 | 26.066 | | Elite360D [1] | S → S | 0.223 | 66.031 | 51.764 |
| | Elite360D [1] | M → M | **0.291** | **67.189** | **54.663** | (b) | SliceNet [43] | M → S | 1.205 | 21.681 | 12.763 |
| | Depth Anywhere [60] | M$^+$ → M | 0.420 | 53.997 | 39.482 | | UniFuse [27] | M → S | 0.277 | 56.150 | 40.678 |
| (b) | SliceNet [43] | S → M | 0.873 | 24.433 | 14.734 | | ACDNet [77] | M → S | 0.495 | 43.497 | 28.586 |
| | UniFuse [27] | S → M | 0.542 | 33.832 | **21.711** | | BiFuse++ [59] | M → S | 0.770 | 46.650 | 31.731 |
| | ACDNet [77] | S → M | 0.580 | 32.277 | 19.985 | | Elite360D [1] | M → S | **0.184** | **73.501** | **60.175** |
| | Elite360D [1] | S → M | **0.539** | **34.489** | 21.548 | | Depth Anywhere [60] | M$^+$ → S | 0.498 | 50.927 | 35.021 |
| (c) | 360MonoDepth [47] | 🤖 → M | 0.632 | 28.056 | 16.998 | (c) | 360MonoDepth [47] | 🤖 → S | 0.576 | 25.230 | 14.672 |
| | RPG360 (Ours) | 🤖 → M | **0.337** | **58.816** | **44.912** | | RPG360 (Ours) | 🤖 → S | **0.206** | **73.917** | **61.880** |

structural awareness for practical applications [52, 40], we adopt 3D metrics, such as Chamfer Distance, F-Score and IoU, rather than 2D metrics to better assess improvements in 3D structure and geometry. Further details of the 3D metrics are described in the supplementary materials.

Table 2: Quantitative comparison on 360Loc (**L**). For Depth Anywhere, we train and test on 360Loc using its pseudo labeling (**L$_p$**) technique. Among optimization-based methods leveraging perspective foundation models (🤖), RPG360 improves the Chamfer distance by 17.2% compared to 360MonoDepth.

| | Method | Dataset Train → Test | 3D metrics Chamfer ↓ | F-Score ↑ | IoU ↑ |
|---|---|---|---|---|---|
| (a) | SliceNet [43] | M → L | 2.941 | 8.164 | 4.409 |
| | UniFuse [27] | M → L | 2.533 | 10.373 | 5.670 |
| | ACDNet [77] | M → L | 2.423 | 10.572 | 5.728 |
| | BiFuse++ [59] | M → L | 2.379 | 8.682 | 4.647 |
| | Elite360D [1] | M → L | 1.759 | 14.797 | 8.565 |
| | Depth Anywhere [60] | M$^+$ → L | 2.226 | 11.889 | 6.571 |
| | Depth Anywhere [60] | M$^+$, L$_p$ → L | **1.457** | **17.145** | **10.112** |
| (b) | 360MonoDepth [47] | 🤖 → L | 2.274 | 8.992 | 4.873 |
| | RPG360 (Ours) | 🤖 → L | **1.883** | **19.605** | **11.502** |

**Comparison Models**  As also noted in [60], many recent methods [1, 2, 73, 36, 37, 51, 72] have not fully released their pre-trained models or provided their code and implementation details. Some also use different datasets, making direct comparison difficult. Therefore, we evaluate our method on fully accessible benchmarks, comparing it with both learning-based [43, 27, 77, 59, 1, 60] and optimization-based [47] approaches using 3D metrics. In addition, to demonstrate the effectiveness of our approach in relation to recent methods, we also include evaluations on 2D metrics. For each method, We adopt the backbone or pre-trained model that achieved the highest performance as reported in the original paper (e.g., EfficientNet-B5 [55] for Elite360D [1] and BiFuse++ for Depth Anywhere [59]). To mitigate scale ambiguity in monocular depth estimation, we apply median alignment [76] in all evaluations. Further details are provided in the supplementary material.

**Implementation Details**  We use the Adam optimizer [33] to perform gradient descent on a single RTX A5000 GPU. To accelerate convergence, we adopt a multi-scale approach following [50]. The ERP depth map $D^{ERP^l} \in \mathbb{R}^{\lfloor h/2^l \rfloor \times \lfloor w/2^l \rfloor}$ where $l \in \{0, ..., L - 1\}$, is downsampled by a factor of $2^l$. In this experiment, we set $L = 3$ and use learning rates of $5 \times 10^{l-L}$. Each optimization step is performed for 300, 150, and 30 iterations. The weights of the loss terms $\eta_p$, $\eta_d$, and $\eta_n$ are set to 50, 0.5, and 10, respectively.

### 4.2 Experimental Results

As shown in Table 1, although supervised methods perform well when the training and testing domains are the same, their performance degrades significantly when applied to different domains. Interestingly, Table 1 also reveals the opposite trend: some of the methods, such as Elite360D [1], trained in Matterport3D achieve better performance on the Stanford2D3D test set than those trained on the Stanford2D3D train set. Given that the Matterport3D dataset is much larger than Stanford2D3D, we can infer that the available volume of labeled 360° data in Stanford2D3D might be insufficient for effective learning, depending on network capacity. This demonstrates that the lack of labeled 360° datasets poses a significant challenge for supervised learning approaches.

As a potential solution to this problem, the perspective foundation models offer advantages because of their inherent robustness across diverse scenes, and they leverage knowledge from large-scale datasets. 360MonoDepth [47], which employs the perspective foundation model with a blending algorithm,

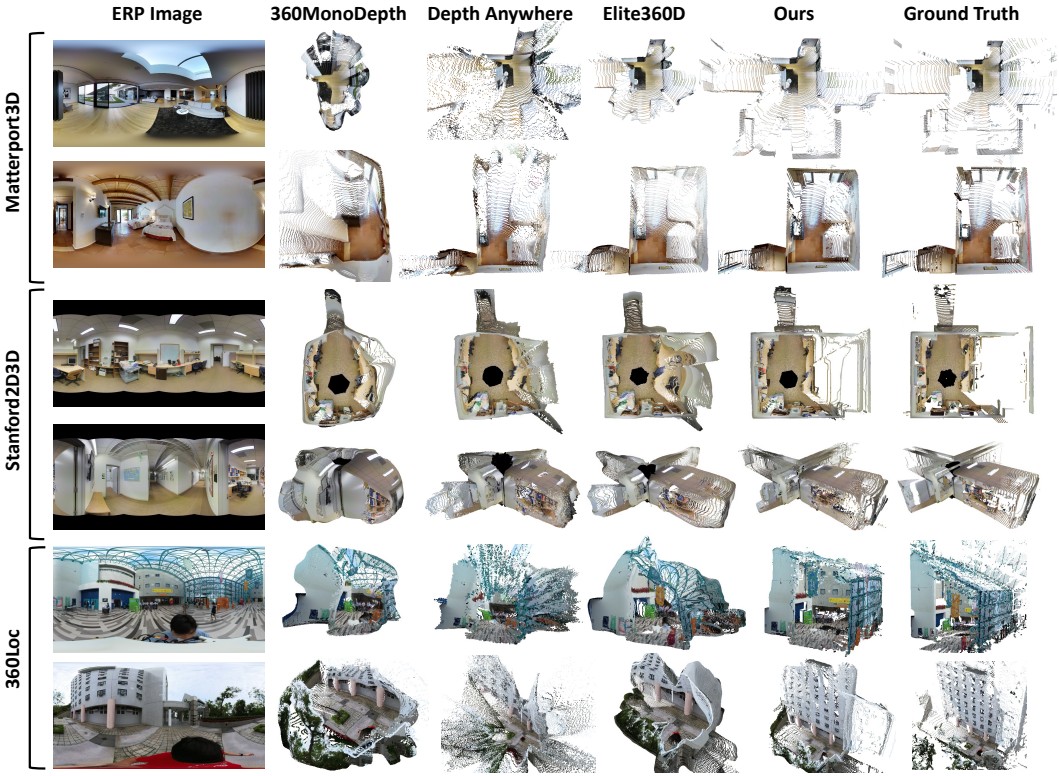

Figure 4: Qualitative comparison of 3D point clouds reconstructed from depth maps across different datasets. Depth Anywhere and Elite360D are learning-based methods, while 360MonoDepth and Ours are optimization-based methods with perspective foundation models. Our RPG360 demonstrates superior structural accuracy, most closely matching the ground truth in 3D space.

achieves high-quality depth estimation in 2D-based metrics [47]. However, its performance in 3D metrics is suboptimal, as it ignores 3D structural information during optimization. Depth Anywhere [60] addresses the lack of labeled datasets by generating a pseudo ground truth through a perspective foundation model. While it demonstrates competitive performance compared to other supervised learning methods, its structural information appears distorted when visualized in 3D point clouds, as shown in Fig. 4. By incorporating our graph-based optimization with the additional per-face parameter, we demonstrate that our method achieves comparable or superior performance across diverse datasets. Furthermore, Fig. 4 shows our method produces point clouds with significantly fewer artifacts in 3D space, further validating its effectiveness.

Table 2 presents zero-shot quantitative results on 360Loc, which consists of both indoor and outdoor scenes. For a fair comparison, we trained Depth Anywhere [60] using its pseudo-labeling method on 360Loc and evaluated its performance. Elite360D [1] performs well in indoor scenes like its training domain but exhibits performance degradation in zero-shot tests. Our method achieves better performance in terms of F-Score and IoU, demonstrating its robustness in diverse environments. We conduct 2D metric evaluations on the Matterport3D dataset as shown in Table 3. Learning-based methods generally demonstrate stronger performance on 2D metrics than optimization-based methods. We also evaluate optimization-based methods using high-resolution 2K images, following [47, 42]. To ensure a fair comparison, we implement 360MonoDepth [47] with Metric3D v2 [25]. However, since 360MonoDepth is originally designed to output inverse depth (as in MiDaS v3 [46]), using a model that predicts standard depth results in degraded blending performance. In contrast, our method requires depth predictions. Therefore, we only evaluate methods that directly produce depth outputs [25, 11], in order to avoid additional normalization issues when converting inverse depth to depth. Since Peng and Zhang [42] leverages both a perspective and a 360° network, it outperforms 360MonoDepth, which relies solely on a perspective network. Although our method relies exclusively on a perspective model, it still achieves superior performance, benefiting from both the robustness of foundation models and the structural awareness introduced by graph optimization.

Table 3: Quantitative comparison using 2D metrics on the (a, b) Matterport3D (**M**, 1024 × 512) and (c, d) Matterport3D-2K (**M-2K**, 2048 × 1024) test sets. In the 2K high-resolution setting, (c) learning-based methods (360 Train.) exhibit performance degradation, whereas (d) optimization-based methods (Opti.), including ours, demonstrate stronger performance.

| | Method | Backbone or Pretrained Model | 360 Train. | Opti. | Dataset Train → Test | Lower is better Abs Rel | RMSE | Higher is better $\delta_{1.25}$ | $\delta_{1.25^2}$ | $\delta_{1.25^3}$ |
|---|---|---|---|---|---|---|---|---|---|---|
| (a) | SliceNet [43] | ResNet50 [24] | ✓ | ✗ | M → M | 0.176 | 0.613 | 0.872 | 0.948 | 0.972 |
| | UniFuse [27] | ResNet34 [24] | ✓ | ✗ | M → M | 0.106 | 0.494 | 0.890 | 0.962 | 0.983 |
| | EGFormer [73] | Transformer | ✓ | ✗ | M → M | 0.147 | 0.603 | 0.816 | 0.939 | 0.974 |
| | ACDNet [77] | ResNet50 [24] | ✓ | ✗ | M → M | 0.101 | 0.463 | 0.900 | 0.968 | 0.988 |
| | BiFuse++ [59] | ResNet34 [24] | ✓ | ✗ | M → M | 0.112 | 0.485 | 0.881 | 0.966 | 0.987 |
| | HRDFuse [2] | ResNet34 [24] | ✓ | ✗ | M → M | 0.117 | 0.503 | 0.867 | 0.962 | 0.985 |
| | Elite360D [1] | EfficientNet-B5 [55] | ✓ | ✗ | M → M | 0.105 | 0.452 | 0.899 | 0.971 | **0.991** |
| | Depth Anywhere [60] | BiFuse++ [59], Depth Anything [67] | ✓ | ✗ | M⁺ → M | **0.085** | - | **0.917** | **0.976** | **0.991** |
| (b) | 360MonoDepth [47] | MiDaS v2 [46] | ✗ | ✓ | 🌐 → M | 0.264 | 0.916 | 0.612 | 0.854 | 0.941 |
| | **RPG360** (Ours) | Omnidata v2 [11] | ✗ | ✓ | 🌐 → M | 0.215 | 0.672 | 0.796 | 0.935 | 0.973 |
| | **RPG360** (Ours) | Metric3D v2 [25] | ✗ | ✓ | 🌐 → M | **0.203** | **0.667** | **0.859** | **0.953** | **0.977** |
| (c) | Elite360D [1] | EfficientNet-B5 [55] | ✓ | ✗ | M → M-2K | 0.309 | 0.979 | 0.538 | 0.824 | 0.930 |
| | Depth Anywhere [60] | BiFuse++ [59], Depth Anything [67] | ✓ | ✗ | M⁺ → M-2K | 0.167 | 0.789 | 0.815 | 0.947 | 0.978 |
| | Peng and Zhang [42] | UniFuse [27], LeRes [70] | ✓ | ✓ | M → M-2K | **0.115** | **0.611** | **0.871** | **0.954** | **0.982** |
| (d) | 360MonoDepth [47] | MiDaS v3 [46] | ✗ | ✓ | 🌐 → M-2K | 0.208 | 0.791 | 0.656 | 0.890 | 0.961 |
| | 360MonoDepth [47] | Metric3D v2 [25] | ✗ | ✓ | 🌐 → M-2K | 0.300 | 1.144 | 0.455 | 0.766 | 0.897 |
| | **RPG360** (Ours) | Omnidata v2 [11] | ✗ | ✓ | 🌐 → M-2K | 0.147 | 0.545 | 0.820 | 0.947 | 0.980 |
| | **RPG360** (Ours) | Metric3D v2 [25] | ✗ | ✓ | 🌐 → M-2K | **0.107** | **0.464** | **0.899** | **0.969** | **0.987** |

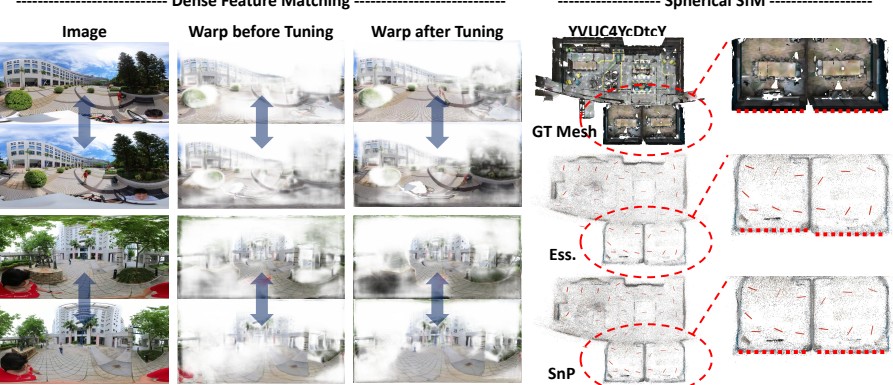

--------------------------- Dense Feature Matching ---------------------------   ------------------ Spherical SfM ------------------

Figure 5: Qualitative results of the downstream tasks using our method. For dense feature matching, EDM [30] estimates more confidential correspondences after tuning with pseudo ground truth generated by RPG360, compared to the initial results. For initial pose estimation in IM360 [31], SnP with RPG360 reconstructs a slightly more accurate 3D structure than two-view geometry-based estimation (Ess.).

## 4.3 Experimental Analysis

To highlight the significance and utility of robust depth estimation, we apply our method to down-stream 360° vision tasks. By leveraging our 360° depth estimation method, we can estimate the relative poses between multiple view frames. To accomplish this, we implement the Spherical-n-Point (SnP) algorithm [21] in conjunction with RANSAC [13].

**Dense Feature Matching**   Since obtaining ground truth correspondences in 360° images is challenging, most existing learning-based feature matching approaches for 360° datasets are primarily restricted to indoor environments [17, 18, 30]. Consequently, when evaluating the feature matching method [30] in outdoor scenes, such as 360Loc, we observe performance degradation. To address this issue, we utilize our robust depth estimation method, which leverages the perspective foundation model. With the aid of SnP and RANSAC, pseudo ground truth correspondences are generated using the relative pose and depth maps from RPG360 in 360Loc query scenes. We then finetune EDM for several iterations and evaluate it on the 360Loc mapping sequences. Table 4 and Figure 5 showcase the advantages of our proposed depth module.

**Structure from Motion**   IM360 [31] demonstrates significant improvements in localization and mapping for large-scale indoor scenes sparsely captured with 360° cameras. During the localization process, relative poses are initialized using two-view geometry estimation based on epipolar geometry and essential matrix decomposition in spherical cameras. Instead of relying on this two-view geometry, we employ SnP [21] with RANSAC [13], utilizing our predicted depth maps before performing spherical incremental triangulation. As shown in Table 6, our approach achieves improvements in AUC errors across most scenes, demonstrating that robust depth estimation can boost the performance of other vision tasks. The qualitative result is presented in Fig. 5.

Table 4: Performance of dense feature matching [30] with and without tuning using RPG360. Our robust depth estimation improves AUC@5° by 5.4% in the hall scene of 360Loc.

| Method | Scene | Tuning | AUC ↑ @5° | @10° | @20° |
|--------|-------|--------|------|------|------|
| EDM [30] | atrium | ✗ | 37.19 | 65.01 | 82.11 |
| EDM [30] | atrium | ✓ | **38.37** | **65.99** | **82.62** |
| EDM [30] | concourse | ✗ | 33.13 | 56.97 | 75.81 |
| EDM [30] | concourse | ✓ | **34.53** | **58.52** | **77.26** |
| EDM [30] | hall | ✗ | 37.83 | 63.20 | 80.92 |
| EDM [30] | hall | ✓ | **39.88** | **64.67** | **81.57** |
| EDM [30] | piatrium | ✗ | 44.45 | 69.91 | 84.86 |
| EDM [30] | piatrium | ✓ | **46.49** | **70.34** | **85.06** |

Table 5: Ablation study on different foundation models and the impact of the per-face parameter $\lambda_c$ on 3D reconstruction performance. While graph optimization alone provides limited performance improvement, incorporating our proposed per-face parameter $\lambda_c$ leads to superior results, highlighting its effectiveness in 3D reconstruction quality.

| Foundation Model | Graph Opti. | Per-face Param. $\lambda_c$ | 3D metrics Chamfer ↓ | F-Score ↑ | IoU ↑ |
|------------------|-------------|------------------------------|-----------------------|-----------|-------|
| Marigold [32] | ✓ | ✓ | 0.916 | 23.962 | 14.262 |
| GeoWizard [15] | ✓ | ✓ | 0.752 | 27.923 | 16.919 |
| Omnidata v2[11] | ✓ | ✓ | 0.402 | 50.035 | 35.968 |
| Metric3D v2[25] | ✗ | ✗ | 0.380 | 51.366 | 36.971 |
| Metric3D v2[25] | ✓ | ✗ | 0.382 | 51.605 | 37.303 |
| Metric3D v2[25] | ✓ | ✓ | **0.337** | **58.816** | **44.912** |

Table 6: Performance of spherical Structure from Motion (SfM) [31] using Epipolar geometry (Ess.) or Spherical-n-Point (SnP) [21] as the initial pose. RPG360 enables the use of SnP, resulting in a 9.7% improvement in AUC@3° for the YVUC3YcDtcY scene in Matterport3D, demonstrating the effectiveness of our robust depth estimation in enhancing spherical SfM performance.

| Method | Scene | Init Pose | AUC ↑ @3° | @5° | @10° |
|--------|-------|-----------|------|------|------|
| IM360 [31] | 2t7WUuJeko7 | Ess. | 49.16 | 69.05 | 84.53 |
| IM360 [31] | 2t7WUuJeko7 | SnP | **51.17** | **70.48** | **85.24** |
| IM360 [31] | 8194nk5LbLH | Ess. | **34.91** | 44.16 | 66.87 |
| IM360 [31] | 8194nk5LbLH | SnP | 34.45 | **46.56** | **67.25** |
| IM360 [31] | pLe4wPe7qrG | Ess. | 73.95 | 84.37 | 92.18 |
| IM360 [31] | pLe4wPe7qrG | SnP | **74.19** | **84.52** | **92.26** |
| IM360 [31] | RPmz2sHmrrY | Ess. | **53.44** | **71.87** | **85.93** |
| IM360 [31] | RPmz2sHmrrY | SnP | 52.92 | 71.54 | 85.77 |
| IM360 [31] | YVUC4YcDtcY | Ess. | 72.40 | 83.40 | 91.71 |
| IM360 [31] | YVUC4YcDtcY | SnP | **79.44** | **87.67** | **93.83** |
| IM360 [31] | zsNo4HB9uLZ | Ess. | 52.35 | 71.14 | 85.49 |
| IM360 [31] | zsNo4HB9uLZ | SnP | **53.50** | **71.84** | **85.86** |

**Without Scale Param. λ**  **With Scale Param. λ**

Figure 6: Effect of the per-face scale parameter $\lambda_c$. (Left) Without $\lambda_c$, Eq. 6 only smooths the gap between 3D points from different face views within the cubemap, leading to noticeable distortions and misaligned structures. (Right) By incorporating $\lambda_c$ in Eq. 8, depth scale consistency is improved, resulting in enhanced structural integrity.

**Ablation Study**  We analyze the performance variations of our method when using different perspective foundation models, as shown in Table 5. Marigold [32] and GeoWizard [15] utilize Stable Diffusion [48], whereas Omnidata [11] and Metric3D [25] are trained on large-scale datasets using multi-task learning. All these models are capable of predicting both depth and surface normal maps. Although diffusion-based models [32, 15] generate high-fidelity depth maps, their 3D point cloud reconstructions suffer from quality degradation when used for scene reconstruction. Table 5 also demonstrates the effectiveness of the per-face parameter $\lambda_c$ in graph optimization. Without $\lambda_c$, performance remains similar regardless of whether graph optimization is applied. However, incorporating our proposed per-face parameter $\lambda_c$ leads to noticeable improvements, confirming its contribution to reconstruction quality. The corresponding qualitative improvements are illustrated in Fig. 6. The per-face scale parameter $\lambda_c$ provides flexibility for 3D points to move collectively within each face, preventing adjacent points from different faces from being simply normalized. This encourages depth consistency while preserving the original structure.

## 5  Conclusion, Limitations, and Future Work

In this paper, we propose a novel framework for 360° depth estimation. Leveraging foundation models for perspective images, we convert a single ERP image into six-face cubemap images and predict depth and surface normal maps for each face. To ensure depth scale consistency across different faces of the cubemap, we parameterize the predicted depth and normal maps and employ a graph optimization technique with the proposed per-face scale parameter for depth scale alignment. Our optimization-based approach may produce less detailed depth estimations for thin structures compared to training-based methods. However, by leveraging prior knowledge from perspective foundation models, it demonstrates strong robustness in zero-shot settings. Furthermore, as more powerful foundation models emerge, we expect the advantages of our method to become even more pronounced. In the future, we aim to extend the robustness of our approach beyond SfM to applications such as multiview image reconstruction.

**Acknowledgements** This work was supported in part by ARO Grant W911NF2310352 and Army Cooperative Agreement W911NF2120076.

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
