# OpenReview forum: "RPG360: Robust 360 Depth Estimation with Perspective Foundation Models and Graph Optimization"
_NeurIPS.cc/2025/Conference — NeurIPS 2025 poster_

### Official Review · Reviewer_PqBQ · 2025-06-28

**Clarity:** 3
**Significance:** 3
**Originality:** 2
**Rating:** 3
**Confidence:** 4

**Summary:**

This paper introduces RPG360, a novel training-free method for monocular 360-degree depth estimation. The core idea is to leverage the robust capabilities of pre-trained perspective foundation models, circumventing the need for large, labeled 360-degree datasets. The approach begins by converting an equirectangular (ERP) image into a six-face cubemap representation, where each face is processed independently by a perspective foundation model to predict initial depth and surface normal maps. The central contribution lies in a graph-based optimization framework designed to address the scale inconsistencies that arise when merging these independently estimated depth maps from different cubemap faces. This optimization refines the depth and normal maps by incorporating a per-face scale parameter, ensuring global depth scale consistency and preserving 3D structural integrity. The authors demonstrate the method's effectiveness and robustness in zero-shot settings across diverse datasets like Matterport3D, Stanford2D3D, and 360Loc, and illustrate its utility in downstream tasks such as feature matching and Structure-from-Motion.

**Questions:**

1. Specificity of Foundation Model Version: In Ln. 276, the paper references "Metric3D v2 [25]". Could the authors confirm that all instances where Metric3D is utilized as the foundation model throughout the paper, particularly for the "RPG360 (Ours)" results in Tables 1, 2, and 3, consistently refer to and use "Metric3D v2" as detailed in reference [25]?

2. Could the authors reformat Table 4. and Table 5. for readability:
The presentation of quantitative results in Table 4 and Table 5 would benefit from enhanced readability. We suggest the authors consider reformatting these tables to improve clarity, such as by adjusting column spacing, ensuring consistent value alignment, or utilizing more distinct visual separations between logical sections. This would facilitate easier interpretation of the presented data.

3. Comparison with Planar-Based 360 and 3D Reconstruction Methods:
Given the paper's emphasis on 3D metrics and structural correctness in depth estimation, a comparison with established planar-based 3D reconstruction methods would provide valuable context. These methods also prioritize 3D alignment and geometric accuracy. Could the authors include a comparative analysis against representative planar-based approaches to further contextualize RPG360's performance in 3D structural awareness?

4. Inference Time and Computational Efficiency:
The paper currently lacks a comprehensive analysis of inference time. To fully assess the practical applicability and computational efficiency of RPG360, please provide the end-to-end processing time for a single 360-degree image. This should include the time taken for depth estimation by the foundation model and the subsequent graph optimization step.

5. Robustness to Image Degradation:
The paper highlights the robustness of RPG360. While zero-shot generalization to diverse scenes is demonstrated, it is crucial to understand the method's performance under various image degradations (e.g., low-light conditions, noise, motion blur). Given that the initial depth and normal maps from perspective foundation models may degrade in such scenarios, how does the proposed graph optimization specifically perform and affect the final depth quality under these challenging input conditions? Does it exhibit graceful degradation, amplify errors, or provide resilience?

6. Limitations in Thin Structures and Fine Details: In Section 5, the authors mention, 'Our optimization-based approach may produce less detailed depth estimations for thin structures compared to training-based methods.' Could the authors expand on the specific reasons for this limitation? Is it inherent to the graph optimization's smoothing properties, limitations of the cubemap projection near edges, or a characteristic inherited from the foundation models? Discussing potential strategies or future work to mitigate this limitation would enhance the paper's completeness.

7. Visualizations of Failure Modes/Limitations: Beyond quantitative metrics, qualitative visualizations of the method's limitations or failure modes would be highly beneficial. Specifically, could the authors provide examples (e.g., in supplementary material) illustrating scenarios where 'less detailed depth estimations for thin structures' are evident, or where distortions or inconsistencies persist despite the graph optimization? This would offer crucial insights into the method's current boundaries.

**Ethical Concerns:**

["NO or VERY MINOR ethics concerns only"]

**Final Justification:**

### Justification for Final Decision

After reviewing the authors' rebuttal, my initial concerns regarding the paper's scientific rigor remain, as the explanations were not persuasive on key issues of originality, experimental fairness, and robustness.

---
### Addressed Points
I would like to acknowledge that the authors have committed to updating the model specifications and adding visualizations of limitations, which will improve the paper's clarity. The new robustness validation against image degradation is a welcome addition, as is the detailed explanation for the "thin structures" limitation (though this would be more convincing with direct experimental support).

---
### Originality Claim and Literature Gap

The authors' originality claim is overstated. While their specific combination of a training-free approach with graph optimization is novel, the fundamental concept of incorporating structural correctness (e.g., via planar priors or structural losses) is not, as shown by prior work [1, 2]. The paper mischaracterizes the existing literature by not acknowledging these precedents.

---
### Fairness of Comparison

The experimental design has significant issues that undermine the paper's conclusions.

* **Unbalanced Baselines:** A key premise of this work is that leveraging a massive perspective foundation model should lead to superior generalization in zero-shot scenarios where supervised methods struggle. However, the evaluation shows the proposed method achieves only similar results to the state-of-the-art Elite360D[4] on the Stanford2D3D dataset (when the baseline is trained on Matterport3D). This lack of a significant performance gap, despite the massive data advantage of the foundation model, makes it difficult to ascertain the true effectiveness of the proposed graph optimization.

* **Invalid Justification:** The rebuttal on this point was particularly concerning. While the authors state in their response that they "included both the original performance of Depth Anywhere and additional results using Metric3D V2 to allow for better contextualization," these specific results are not present in the submitted paper nor rebuttal. For a meaningful comparison between methods that both leverage perspective foundation models, the underlying models must be aligned. Justifying an unfair comparison by referencing non-existent data does not resolve the core methodological issue.

---
### Conclusion

In conclusion, while I acknowledge the effort in the rebuttal and the new experiments provided, the core concerns regarding the paper's positioning on novelty and the fairness of key experimental comparisons remain. Because these issues significantly impact the confidence in the paper's conclusions, I have adjusted my score from Borderline Accept to Borderline Reject.

***
[1] Indoor Panorama Planar 3D Reconstruction via Divide and Conquer\
[2] Pano3D: A Holistic Benchmark and a Solid Baseline for 360° Depth Estimation\
[3] Enhancing 360 Monocular Depth Estimation via Perspective Distillation and Unlabeled Data Augmentation\
[4] Hao Ai and Lin Wang. Elite360d: Towards efficient 360 depth estimation via semantic-and distance301 aware bi-projection fusion.

**Limitations:**

yes

**Quality:**

3

**Strengths And Weaknesses:**

### Strengths of RPG360

* **Effective Adaptation of Perspective Foundation Models**: The paper presents an innovative strategy for leveraging powerful, pre-trained perspective depth models for 360-degree imagery. This "training-free" approach is a significant advantage, as it inherently bypasses the challenges of limited large-scale, labeled 360-degree datasets and mitigates domain generalization issues commonly faced by supervised learning methods in this domain.
* **Novel Graph Optimization for Consistent Scale Alignment**: A key technical strength is the introduction of a novel graph-based optimization framework, notably incorporating a per-face scale parameter ($\lambda_c$). This method offers an effective solution to the critical problem of depth scale inconsistencies that arise when fusing independently estimated depth maps from multiple cubemap faces. The ablation study (Table 5) clearly demonstrates the substantial positive impact of this per-face parameter on the quality of 3D reconstruction.
* **Robustness and Strong Zero-Shot Generalization**: RPG360 consistently exhibits robust performance in zero-shot settings across a variety of datasets, including Matterport3D, Stanford2D3D, and 360Loc. This is a notable benefit, as the method effectively operates without requiring 360-degree training datasets and is consequently less susceptible to domain gaps often encountered by other approaches.
* **Demonstrated Utility in Downstream Applications**: The paper effectively showcases the practical applicability of its high-quality depth maps by validating their benefits in common 360-degree vision tasks such as dense feature matching and Structure-from-Motion (SfM). This provides compelling evidence of the method's real-world impact beyond raw metric scores.

### Weaknesses of RPG360

* **Implicit Foundation Model Specification for Main Results**: While the ablation study (Table 5) lists and evaluates several foundation models, the specific model (including its version and backbone configuration) used to generate the primary "RPG360 (Ours)" results in Tables 1, 2, and 3 is not explicitly stated. For complete clarity and reproducibility, this crucial detail should be directly declared in the respective table captions or accompanying text.
* **Limited Comparative Analysis of Optimization Strategies**: The paper thoroughly describes its proposed graph optimization approach and its internal components. However, it lacks a broader discussion or comparative analysis against other established or alternative global scale alignment or fusion optimization techniques present in the existing literature. This omission makes it challenging to fully assess the distinct advantages and necessity of their particular graph formulation compared to other potential solutions in the domain.
* **Partial Reproducibility Details for Certain Baselines**: Although some baseline models are identified (e.g., "BiFuse++ [59]" for "Depth Anywhere [60]"), comprehensive retraining specifics, such as the exact number of epochs or precise learning rate schedules, are not provided for all models, particularly those that undergo training on pseudo-labeled data (e.g., "Depth Anywhere [60]" on 360Loc). This lack of detail could hinder the exact reproduction of certain baseline results.
* **Lack of Explicit Surface Normal Ablation**: While surface normals are integrated into the graph optimization's formulation (Eq. 6), the paper does not present a dedicated ablation study that directly isolates and quantifies the performance impact of using normals versus entirely omitting them from the optimization framework. The current ablation (Table 5) primarily highlights the substantial impact of the per-face scale parameter, leaving the individual contribution of surface normals less clear.
* **Imbalanced Space Allocation**: While the demonstrations of the method's utility in downstream tasks are valuable and well-executed, the considerable space dedicated to these aspects could potentially be optimized for conciseness. This might allow for more in-depth discussions, additional ablation studies focused directly on the core depth estimation methodology, its theoretical underpinnings, or deeper analyses of failure modes within the main paper.
* **Limited Comparative Scope against 3D-Focused Methods**: While the paper effectively compares against learning-based 360-degree depth estimation methods, the absence of a direct comparison with established planar-based 3D reconstruction techniques, which also prioritize geometric alignment and structural correctness, represents a missed opportunity to fully contextualize RPG360's claimed superior 3D structural awareness.
* **Unanalyzed Robustness to Image Degradations**: Although the paper claims "robustness" and demonstrates strong zero-shot generalization to diverse scenes, it does not provide an analysis of the method's performance when presented with common image degradations such as low-light conditions, noise, or blur. It remains unclear how the proposed graph optimization, which relies on initial depth and normal maps, performs or is affected under these challenging input scenarios.
* **Suboptimal Table Formatting**: The visual presentation and separation of information within Tables 4 and 5 are suboptimal. The uneven or off-center alignment of elements within these tables can be distracting and hinder quick interpretation of the presented quantitative results.

---

> ### Author Rebuttal · Authors · 2025-07-29
>
> We appreciate your time and careful evaluation of our work.
> We are glad that the effectiveness of our training-free approach and the core contribution of our graph-based optimization framework with per-face scale alignment were well recognized.
> We also appreciate the recognition of our method’s zero-shot generalization and its utility in downstream applications.
>
> ## **W1Q1) Model Specification**
> In **Table 3** in the main paper, we explicitly indicate the backbone or pretrained model used for each method, including Metric3D V2 and Omnidata V2 for RPG360.
> For the main results in Tables 1 and 2, RPG360 (Ours) consistently uses Metric3D V2 as the foundation model.
> We will revise the manuscript to explicitly state the foundation model used in Tables 1 and 2 to ensure clarity and reproducibility.
>
> ## **W2Q3) Analysis of Optimization Strategies**
> To the best of our knowledge, there are no existing methods that apply graph-based optimization specifically for 360 depth estimation.
> Among prior works, 360MonoDepth [47] and Peng and Zhang [42] incorporate optimization techniques, but their approaches are limited to blending-based depth alignment without explicit consideration of structural consistency or global 3D geometry (**Lines 47–51**).
> In contrast, our method introduces a graph optimization framework that leverages both depth and surface normals to enforce local planarity and global structural coherence, resulting in more accurate 3D reconstruction.
> Moreover, we analyze the effectiveness of our proposed optimization strategy through an ablation study presented in Table 5.
>
> ## **W3) Reproducibility Details for Depth Anywhere [60]**
> For models that we reimplemented, we followed the original codebase and training configurations as closely as possible to ensure fair comparison.
> Specifically, for Depth Anywhere, we used its best-performing backbone BiFuse++ (**Line 205**), and followed the original settings: training for 200 epochs with a batch size of 4, a learning rate of 3e-4, and the Adam optimizer.
> We selected the best-performing checkpoint based on validation performance for testing.
> We will include these training details in the supplementary material to enhance clarity and reproducibility.
>
> ## **W4) Lack of Surface Normal Ablation**
> In our proposed optimization, surface normals are an essential component and not an optional module.
> Removing normals would effectively mean disabling the core of our method.
> The performance in this setting, without applying our optimization framework, is reported in Table 5, fourth row (Chamfer = 0.380).
> Furthermore, while surface normals can be predicted using a separate network, they can also be computed directly from the predicted depth.
> Thus, incorporating normals does not introduce additional inference overhead, and we do not consider their use a burden to the overall method.
>
> ## **W5) Space Allocation**
> Our primary contribution lies in proposing a robust 360 depth estimation method using a perspective foundation model.
> Given this, we believe it is important to demonstrate not only why robustness matters, but also how the proposed method can be meaningfully applied to various 360 vision tasks.
> These downstream tasks help validate the generalization capability of our method and provide empirical insight into its practical effectiveness.
> Therefore, we allocated a portion of the main paper to downstream applications, which we believe contribute to a more comprehensive understanding of the methodology.
>
> ## **W6Q3) Limited Comparison**
> As noted in our response to W2, our work is the first to explicitly incorporate structural correctness and local planar approximation into a 360 depth estimation framework.
> Instead, we compared our approach with prior optimization-based blending algorithms such as [42, 47].
> We believe this comparison supports the validity and advantages of our graph-based optimization approach.
>
> To the best of our knowledge, most planar-based 3D reconstruction techniques fall under the category of multi-view stereo (MVS), which operates under the assumption of overlapping views.
> In contrast, our graph optimization method is designed to operate on adjacent but non-overlapping cubemap faces in a single 360 image.
> Therefore, these MVS-based approaches are not directly applicable to our problem setting.
>
> If the reviewer was referring to other specific planar-based techniques beyond MVS, we would be happy to consider them and include a discussion in the supplementary material.
>
> ## **W7Q5) Image Degradation**
> Our primary focus in this work is to explore how robust 360 depth estimation can be achieved in the absence of large-scale 360 training datasets, across both indoor and outdoor scenes (**Line 60**).
> While we agree that evaluating robustness under image degradation such as low-light, noise or motion blur would provide additional insights, acquiring 360 datasets with controlled degradations is challenging.
>
> Also, to ensure fair comparison, we conducted experiments using standard benchmark datasets in 360 depth estimation: Matterport3D [5], Stanford2D3D [4], and recently released 360Loc [26].
> Although they may not frequently contain the specific types of degradations suggested, their diversity in scene types allows us to test generalization across varied conditions.
> Furthermore, because our method builds upon a foundation model trained on large-scale perspective data with strong generalization capabilities, we expect it to remain relatively robust to common image degradations, especially compared to models trained solely on limited 360 data.
>
> ## **W8Q2) Table Formatting**
> Thank you for the helpful suggestion.
> Due to differences in caption length and table layout, aligning and visually separating Tables 4 and 5 precisely at the center was challenging in the current submission.
> We agree that improving visual clarity is important, and we will reformat the tables to enhance readability.
>
> ## **Q4) Inference Time**
> The time required for depth estimation by the foundation model depends entirely on the specific model used.
> The subsequent graph optimization stage introduces additional computation, and its runtime varies depending on the number of optimization iterations.
> We have included a runtime versus performance trade-off **Table** in our response to **Reviewer s1NL’s Q1**.
> With the parameters used in our main experiments, the proposed optimization takes approximately 4.6 seconds per image based on our Python implementation.
> If real-time performance is desired, further speed-ups can be achieved by reducing the number of iterations.
>
> ## **Q6) Thin Structures and Fine Details**
> Learning-based methods often preserve edge-level details better because they directly regress depth from image features, although the resulting 3D points may lack geometric accuracy.
> In contrast, our optimization-based approach focuses on enforcing geometric consistency, but may underperform in preserving very thin structures due to the smoothing nature of graph optimization.
> This limitation could potentially be mitigated by incorporating additional priors, such as semantic segmentation, to selectively break graph connections near object boundaries.
> However, this would increase computational complexity and was considered beyond the scope of this work.
>
> ## **Q7) Visualization of Limitation**
> In some cases, such as thin structures like chair legs that occupy only a small number of pixels, our optimization may smooth out the depth and reduce detail.
> To help clarify this limitation, we will add qualitative examples of such failure cases in the supplementary material.

---

> > ### Comment · Reviewer_PqBQ · 2025-08-03
> >
> > Thank you for the detailed rebuttal. After careful consideration of your responses and the submitted paper, my concerns regarding the originality claims, fairness of comparisons, and overall robustness of the method remain. I appreciate the effort you've put into addressing my questions, but the provided explanations are not persuasive.
> >
> > **Originality Claim and Literature Gap**
> >
> > In your response (W2Q3, W6Q3), you claim to be "the first to explicitly incorporate structural correctness and local planar approximation into a 360 depth estimation framework". However, this is not accurate. The broader concept of enforcing structural correctness for improved 3D reconstruction is not novel in this field. For instance, [1] explicitly addresses integrating structural correctness and planar priors into 360-degree depth estimation. The Pano3D benchmark [2] itself proposed a baseline model leveraging structural losses. While your method may be the first to combine a training-free approach with foundation depth models and your specific graph optimization with structural incorporation, the claim of being the first to incorporate structural correctness needs to be contextualized with existing literature. The paper would be stronger if it acknowledged this prior work and focused on the novelty of its specific combination of components.
> >
> > **Fairness of Comparison**
> >
> > As Reviewer vRvv's W2Q2 noted, the paper claims superior performance on Matterport3D in a zero-shot setting. However, the state-of-the-art baseline, Elite360D, was trained on a small, homogeneous dataset like Stanford2D3D. This makes it difficult to determine if the performance gain is due to the proposed graph optimization or simply a result of the baseline being disadvantaged by limited training data.
> >
> > Regarding the comparison with Depth Anywhere [3], I also find the rebuttal unconvincing. Since Depth Anywhere is a distillation technique that leverages a foundation model to generate pseudo ground truth to deal with lack of 360 labeled data, a fair comparison requires a consistent foundation model setting (this is already neglecting the mention of SOTA 360 backbone is interchangeable in the paper). Aligning the foundation model used for both methods is crucial to isolate the performance difference to the core methodologies and not to the choice of the underlying model. The current comparisons may not be fair, as they could be reflecting differences in the underlying foundation models rather than the effectiveness of your proposed graph optimization.
> >
> > **Model Robustness and Evaluation Gaps**
> >
> > I maintain my concern that a proper analysis of robustness against image degradations is still missing. The paper's claim of "robustness" is not fully supported by the evaluation, as it lacks tests for in-the-wild scenarios. I also agree with Reviewer vRvv that the model's performance should be tested in more diverse scenarios such as outdoor scenes.
> >
> > **Conclusion**
> >
> > In conclusion, while I acknowledge the effort in the rebuttal, the core technical concerns regarding the novelty claim, the fairness of the experimental comparisons, and the lack of a proper robustness analysis remain unaddressed. These issues are significant enough to impact my final assessment of the paper's scientific rigor and overall quality. My original score was base on question and weakness well targeted, and I'm downgrading my score from borderline accept to borderline reject.
> >
> > ***
> > [1] Indoor Panorama Planar 3D Reconstruction via Divide and Conquer \
> > [2] Pano3D: A Holistic Benchmark and a Solid Baseline for 360° Depth Estimation \
> > [3] Enhancing 360 Monocular Depth Estimation via Perspective Distillation and Unlabeled Data Augmentation

---

> > > ### Author Response · Authors · 2025-08-04
> > >
> > > ## **Originality Claim and Literature Gap**
> > > We would like to clarify the key distinctions between our method and the two prior works.
> > >
> > > [1] focuses on instance segmentation using horizon and vertical priors to identify vertically aligned or parallel planes. Its primary goal is plane detection and reconstruction, not depth estimation. It explicitly discards fine-grained depth details, making its objective fundamentally different from ours.
> > >
> > > [2] introduces the Pano3D dataset and employs Virtual Normal Loss, which computes differences between predicted and GT surface normals using randomly sampled 3D points. This loss requires GT depth and is applied during training.
> > >
> > > In contrast, our method is training-free and performs direct optimization at inference time. To our knowledge, this is the first approach to leverage normal cues in a test-time optimization, offering a distinct contribution compared to the learning-based frameworks of [1] and [2].
> > >
> > > [1] focuses on plane estimation and [2] is a dataset proposal.
> > > Therefore, directly comparing our method with these works is not appropriate. Prior 360 depth estimation works (e.g., Elite360D, Depth Anywhere) also do not report quantitative comparisons against [1] or [2].
> > >
> > > Our work is better situated among existing approaches where dense depth prediction itself is the primary goal.
> > > We hope this clarifies the differences and contributions of our approach.
> > >
> > >
> > > ## **Fairness of Comparison**
> > > We would like to clarify several points regarding our evaluation and comparisons: First, we conducted extensive validation not only via cross-validation but also by testing on multiple datasets in consistent settings.
> > >
> > > Given the limited availability of GT 360 depth datasets, large-scale supervised training is inherently constrained.
> > > Furthermore, in real-world scenarios, models often face unseen environments.
> > > In such cases, training-based methods like Elite360D are more prone to generalization issues.
> > >
> > > Although Elite360D reports good quantitative performance, our qualitative analysis (e.g., Fig. 4) suggests that it does not recover accurate 3D geometry.
> > > Rather, it appears to predict an floor depth well, minimizing averaged numerical error without capturing structural detail.
> > >
> > > Regarding fairness in comparison with Depth Anywhere: our method is designed to work with interchangeable backbones, but it requires models that predict depth, not inverse depth.
> > > This is because our optimization framework uses geometric cues in 3D space, which depend on a depth-to-3D projection.
> > > Inverse depth predictions cannot be reliably converted to depth without access to camera intrinsics or other constraints, often unavailable in in-the-wild settings.
> > >
> > > Depth Anywhere, by contrast, is designed to work with inverse depth outputs (e.g., Depth Anything).
> > > Because of this fundamental difference in prediction formats, a direct comparison is not strictly fair.
> > > Nevertheless, we believe comparing results is still valuable and transparent.
> > > Hence, we included both the original performance of Depth Anywhere and additional results using Metric3D V2 to allow for better contextualization.
> > >
> > > Also, as the reviewer pointed out, Depth Anywhere adopts a distillation technique, whereas our method is based on optimization. These two methods pursue fundamentally different directions: our approach introduces a novel way to leverage foundation models through optimization for 360 depth estimation. Therefore, rather than focusing on a direct comparison, we believe our method makes a meaningful contribution by demonstrating an optimization-based pathway in this research area.
> > >
> > > ## **Model Robustness and Evaluation Gaps**
> > > Since no existing 360 benchmark includes degraded images, we conducted an additional evaluation to test robustness under image degradation, using the following factors:
> > > brightness = 0.6, contrast = 1.2, saturation = 0.8, gamma = 1.1, hue = 0.05, and Gaussian noise std = 0.1.
> > >
> > > |name|degradation|a1↑| chamfer↓|
> > > |:--:|:--:|:--:|:--:|
> > > |elite360d|x|0.899 |0.291|
> > > |elite360d |O|0.718|0.513|
> > > |rpg360|x|0.859|0.337|
> > > |rgb360|O|0.843|0.352|
> > >
> > > While Elite360D performs well when trained and tested on Matterport3D, it significantly degrades under image perturbations. In contrast, our method remains robust under the same conditions, demonstrating stronger resilience.
> > >
> > > 360Loc includes outdoor scenes, and we report the corresponding results in Fig. 4 and Tab. 2. Regarding Reviewer **vRvv**'s comments, Dur360BEV, we would like to clarify that this research was published after our submission. As Dur360BEV is the first autonomous driving dataset with spherical imagery, we relied on 360Loc to evaluate performance on outdoor environments.
> > >
> > > We would appreciate it if you could also take a moment to read our response to Reviewer **vRvv**’s comment on this matter.

---

> > > > ### Comment · Reviewer_PqBQ · 2025-08-05
> > > >
> > > > Thank you for the detailed rebuttal. After careful consideration of your responses, I find that my core concerns regarding the paper's claims and evaluation remain unaddressed. While I appreciate the effort, the explanations provided were not persuasive.
> > > >
> > > > **Originality Claim and Literature Gap**
> > > > The authors appear to sidestep the claim of "the first to explicitly incorporate structural correctness and local planar approximation into a 360 depth estimation framework" but focus on comparing [1] and [2] with the work. [1] and [2] may differ, but the novelty of "incorporate structural correctness and local planar approximation into a 360 depth estimation" is shared. While [1]'s primary goal may be plane detection, its method of using "structural correctness" to aid in 3D reconstruction with an intermediate depth estimation module incorporating planar parameters is highly relevant and a direct counterexample to the claim that no prior work incorporates this concept. [2] is indeed a dataset paper, but the baseline it proposes—which uses a structural loss—still serves as a precedent for the general idea of leveraging geometric priors in a learning framework.
> > > >
> > > > **Fairness of Comparison**
> > > > The author claims outperforming on other metrics. However, Elite360D is still competitive with single small 360 dataset pretrained and apply in zero-shot setting. whereas the proposed method leverages an extensively larger pretrained foundation model. Which make this statement weak. As the author claims to be the first to apply zero-shot graph-base optimization on perspective foundation model to deal with "limited data" on 360. It is indeed important to compare to other methods that deals with "limited data" though different method. and inverse depth is still comparable with either applying scaling or with scale-invariant metrics.
> > > >
> > > > **Robustness**
> > > > Thank you for providing a useful ablation study on image deflation. It is a significant improvement on this literal work. Just be careful of the typo in the table.

---

### Official Review · Reviewer_vRvv · 2025-07-02

**Clarity:** 3
**Significance:** 2
**Originality:** 2
**Rating:** 4
**Confidence:** 4

**Summary:**

The authors propose a method that offers a training-free 360° depth estimation method. They leverage perspective foundation models that predict depth for cubemap representations of the 360° scene. The predicted depth maps are aligned through graph-based optimization addressing scale inconsistencies. The method shows superior zero-shot performance compared to training-free foundation models on datasets like Materport3D, Stanford2D3D and 360Loc.

**Questions:**

Could the authors clarify whether the model is intended for real-time deployment or for generating pseudo-ground truth, and provide runtime measurements to support this?

Given that RPG360 is pre-trained on a large and diverse dataset corpus, whereas competing methods like Elite360D are trained only on Stanford2D3D—which you note may be too small to generalize well—have the authors considered whether training such methods on more data (excluding Matterport, which is used only for zero-shot evaluation) would significantly improve their performance on Matterport? Without this comparison, it remains unclear whether RPG360’s superior zero-shot performance stems from its architecture or simply from access to more training data.

Could the authors evaluate the model’s performance on challenging outdoor 360° datasets such as Dur360BEV or scenes to demonstrate its generalization beyond indoor environments where surface assumptions are valid only to a limited extent?

Could the authors provide ablation studies showing the trade-off between the number of optimization iterations and performance, as well as the specific impact of incorporating surface normals?

Why did the authors choose to reimplement 360MonoDepth with a different backbone (Metric3D V2) instead of evaluating your method using 360MonoDepth's original foundation model (e.g., MiDaS V3) and standard 2D metrics?

Can the authors demonstrate whether your method improves existing models like Elite360D when used to generate pseudo-ground truth, in line with approaches such as Depth Anything?

**Ethical Concerns:**

["NO or VERY MINOR ethics concerns only"]

**Final Justification:**

The rebuttal addressed most concerns, including runtime analysis, use case clarity, iteration trade-offs, and design choices. The method is technically sound, well-presented, and useful. Outdoor robustness is only partially evaluated, and pseudo-ground-truth usage is not tested. Therefore, I raise my score to borderline accept.

**Limitations:**

No, runtime is potentially a limitation and should be addressed.

**Paper Formatting Concerns:**

no formatting concerns

**Quality:**

1

**Strengths And Weaknesses:**

Strengths:

The 360-degree foundation model appears useful and relevant.
Figure 4 presents compelling results.
The manuscript is well-written and easy to follow.
The authors plan to release the code, which will benefit the community.

Weaknesses:

A 360-degree foundation model with zero-shot performance is promising, but its value depends on how it can be used. It should be made clear whether the goal is real-time application or generating pseudo-ground truth. Since runtime is not reported, this remains uncertain.

While the zero-shot performance on Matterport is strong, RPG360 lags behind Elite360D by over 10% on Stanford2D3D and on 360Loc. The authors suggest Elite360Ds limited performance on Matterport may be due to Stanford2D3D’s limited training size, which limits generalization for models like Elite360D. However, this comparison seems unfair, as RPG360's foundation models were pre-trained on a much larger and more diverse dataset corpus. It should be examined whether training competing methods such as [1] on more data (excluding Matterport, as it’s used only for zero-shot evaluation) would significantly improve their performance on Matterport. Not addressing this leaves open whether a properly trained alternative could outperform the proposed method as a foundation model.

Unclear how model performs in outdoor environments where surface assumptions (walls etc) don’t apply as much such as indoors -> Dur360BEV (360 autonomous driving dataset) or similar dataset evaluation would be interesting

* Yuan, Chao, et al. "Dur360BEV: A Real-world 360-degree Single Camera Dataset and Benchmark for Bird-Eye View Mapping in Autonomous Driving." ICRA 2025

Ablation studies on both the efficiency trade-off (number of optimization iterations vs. performance) and the impact of incorporating surface normals (and the extent of their improvement) would be helpful.

In line 244ff, the authors note that using a model predicting standard (non-inverse) depth leads to degraded performance in 360MonoDepth. However, their reimplementation uses Metric3D V2, which they acknowledge performs suboptimally. While I agree that 360MonoDepth’s limited adaptability to standard depth values is a drawback, it would be more appropriate to evaluate the proposed method by adapting it, and the evaluation metrics, to 360MonoDepth as-is, rather than reimplementing 360MonoDepth with a different model. A comparison with the proposed method using the foundation model of 360MonoDepth (e.g., MiDaS V3, with surface normals computed via post-processing if needed) and traditional metrics (e.g., AbsRel, MAE, δ₁,₂,₃) would make the reported performance gains more convincing.

It would be valuable to demonstrate that this method can generate high-quality pseudo-ground truth and enhance 360° depth estimation approaches like those in Depth Anything, which rely on perspective-based pseudo labels. For example, evaluating whether zero-shot performance of top-performing models like Elite360D can be further improved using this approach would be interesting.


Minor:

- “For methods not covered in these papers, we referred to the original 30 publications to obtain their reported values.”
Why do the authors mention this? The reader might not have the time to go through all related papers to retrieve reported value. Relevant methods should be included, if not enough space in the main paper they should be included in the supplemental material.

---

> ### Author Rebuttal · Authors · 2025-07-29
>
> We are sincerely grateful for your thoughtful and detailed feedback.
> We are glad that the proposed optimization method leveraging a perspective foundation model was found to be useful and relevant for 360 depth estimation.
> We plan to release the code upon acceptance to further support the community.
>
> ## **W1Q1) Main Purpose and Runtime**
> Our primary goal is not real-time deployment.
> Instead, we aim to develop a robust 360 depth estimation that can serve offline mapping itself, as well as downstream applications such as visual localization (e.g., SfM), as demonstrated in the main paper.
> Due to the lack of large-scale training datasets for 360 images, it is particularly challenging to build a generalized model directly for this modality.
> To address this, we leverage an existing perspective-view foundation model and introduce our proposed optimization technique to enable robust zero-shot 360 depth estimation.
> The downstream tasks are presented to illustrate the importance and utility of such robustness: even though the model is not designed for real-time use, a robust 360 depth estimation can still benefit many 360 vision tasks, such as feature matching and SfM.
>
> ## **W2Q2) Response to Unfair Comparison Concern**
> While RPG360 shows slightly lower Chamfer Distance scores than Elite360D on Stanford2D3D and 360Loc, it actually outperforms Elite360D on other metrics such as F-Score and IoU.
> Moreover, although RPG360 leverages a foundation model pretrained on diverse perspective view datasets, it is not trained on any 360 datasets, unlike Elite360D.
> We acknowledge that training Elite360D on a larger and more diverse corpus (excluding Matterport) could potentially improve its zero-shot performance.
> However, such training assumes access to large-scale 360 training datasets, which is currently unavailable due to the scarcity of 360 ground truth depth data.
> Our goal, instead, is to demonstrate a training-free approach that enables robust 360 depth estimation even under such data-scarce conditions.
> We believe this setup highlights the practical relevance of RPG360 as a viable approach when 360 training data is unavailable or limited.
>
> ## **W3Q3) Performance on Outdoor Dataset**
> Our method is based on local plane approximation at the patch level, which allows it to handle various surface geometries, not only flat structures such as walls but also curved objects, by approximating local tangential planes.
> As a result, it is applicable to both indoor and outdoor scenes, including autonomous driving environments.
> Most prior works on 360 depth estimation adopt indoor datasets such as Matterport3D or Stanford2D3D as standard benchmarks.
> For a fair comparison with existing methods, we followed this convention.
>
> We agree with the reviewer that performance in outdoor environments is important.
> To address this, we additionally evaluated our method on 360Loc [26] (**Line 61**, **Figure 4**), a recently released dataset that includes outdoor scenes.
> Regarding Dur360BEV, we appreciate the reviewer’s suggestion. As this dataset was introduced at ICRA 2025, which took place after our full paper submission, we were not aware of it at the time.
> Nonetheless, we recognize its significance as the first large-scale autonomous driving dataset featuring spherical imagery and plan to include relevant experiments using Dur360BEV in the final version of the paper.
>
> ## **W4Q4) Ablation Study on Iteration and Surface Normal**
> We have added a **Table** summarizing the relationship between the number of iterations and performance in our response to **Reviewer s1NL's Q1**.
> Our results show that performing a sufficient number of iterations at lower resolution, followed by a small number of refinement steps at higher resolution, provides the best trade-off between accuracy and runtime.
> While increasing the number of iterations can further improve performance (type: **double**), we observe that the gains begin to saturate, making the additional runtime less cost-effective.
>
> In our method, surface normals are an essential component for enforcing the local plane approximation and are therefore not an optional module that can be ablated.
> Moreover, surface normals can be either estimated by a separate network or directly computed from the predicted depth, meaning that their use does not require additional supervision or compromise the training-free nature of our approach.
> Therefore, we believe that incorporating normals does not pose a limitation in our framework.
>
> ## **W5Q5) Reimplementation of 360MonoDepth**
> MiDaS V2 and V3, the original backbones used in 360MonoDepth, predict inverse depth.
> However, directly converting inverse depth to standard depth is not straightforward.
> Simply inverting the values does not yield an accurate depth map, as it ignores important factors such as focal length, leading to structural distortions in 3D space.
> Inverse depth represents only a relative inverse distance; thus, naively inverting it does not recover true relative depth or distance.
> As a result, the reconstructed 3D geometry (e.g., point clouds) becomes unreliable, and surface normals computed from such depth maps are also inaccurate.
>
> Since our method relies on the structural integrity of both depth and normals through local plane approximation, using MiDaS V2 or V3 would not result in a meaningful or fair comparison.
> Therefore, we chose to reimplement 360MonoDepth using Metric3D V2, which directly predicts standard depth and preserves structural consistency.
> While Metric3D V2 may not be optimal, we believe it provides a more appropriate and fairer basis for comparison with our method.
>
> ## **W6Q6) Use of Pseudo Ground Truth from RPG360 for Training Elite360D**
> We acknowledge that using RPG360 to generate pseudo-ground truth and further train models like Elite360D would be valuable.
> However, such an approach is inherently limited by the quality of the generated pseudo-ground truth.
> In other words, the maximum achievable performance is bounded by the accuracy of the pseudo labels themselves.
>
> In contrast to Depth Anywhere [60], which rely on pseudo-label training from perspective foundation models, our goal is to demonstrate that a perspective foundation model itself can directly perform robust 360 depth estimation in a zero-shot manner when combined with our proposed optimization technique.
> We believe this highlights a distinct and meaningful contribution that goes beyond merely serving as a pseudo-label generator.
>
> ## **W-minor)**
> To clarify, the sentence means that we included the performance of all relevant methods in our tables.
> In 360 depth estimation research, the same methods are often reported with slightly different performance values across different papers, which can lead to confusion.
> To ensure consistency and fairness, we followed the values reported in the most recent and comprehensive benchmark papers [1] and [18] when available.
> For methods not covered in [1] or [18], we referred to their original papers to obtain the reported performance.
> Our intention was to avoid inconsistencies in reported metrics, not to imply that any prior works were excluded from our comparison.
> We will revise the wording in the final version to make this point clearer and will include the full set of reference values in the supplementary material for transparency.

---

> > ### Comment · Reviewer_vRvv · 2025-08-08
> >
> > I appreciate the authors’ thorough runtime analysis and believe this is an important aspect to discuss openly in the final version of the paper. I also value the detailed answers provided in the rebuttal, especially the clarifications on statements such as “directly perform robust 360 depth estimation in a zero-shot manner when combined with our proposed optimization technique” and the broader motivation beyond limited 360 training data — namely, that the method can serve offline mapping as well as downstream applications like visual localization.
> >
> > While I understand and acknowledge the authors’ reasoning for not including certain benchmark comparisons, I remain somewhat unconvinced that the optimization-based method will generalize well in outdoor scenarios as a foundation model. It would be valuable to see experiments on autonomous driving sequences, and, where the system encounters limitations (e.g., low light, adverse weather, degraded image quality), to have these openly discussed. This would not diminish the work but rather highlight directions for future research. I hope the authors keep their promise for experiments on Dur360BEV.

---

> > > ### Author Response · Authors · 2025-08-09
> > >
> > > Thank you for your thoughtful comments on our method’s applications and runtime evaluation.
> > >
> > > Following Reviewer **PqBQ**’s suggestion, we will add image degradation results, as shown in the **Model Robustness and Evaluation Gaps** Table below.
> > >
> > > Although Dur360BEV was published after our submission, we recognize the importance of this comparison and will include additional Dur360BEV results in the supplementary material of the final version.
> > >
> > > We believe these additions will directly address your concern and further clarify the method’s applicability.

---

### Official Review · Reviewer_78uu · 2025-07-03

**Clarity:** 3
**Significance:** 3
**Originality:** 2
**Rating:** 4
**Confidence:** 3

**Summary:**

The paper proposes RPG360, a training-free pipeline for monocular 360-degree depth estimation. A panorama is converted to a 6-face cubemap; each face is fed to an off-the-shelf perspective foundation model to predict depth and normals. A subsequent graph optimisation introduces a learnable per-face scale λ_c and planar depth–normal constraints, aligning all faces into a single, scale-consistent ERP depth map.

**Questions:**

1. Considering that graph optimization may involve complex iterative calculations and may be limited by computational resources in practical applications.  Please report average optimisation time and GPU/CPU memory per 1024×512 panorama.  A plot of objective vs iterations would help.
2. How does the optimization of the per-face scale parameter λc in Eq 8 prevent depth discontinuity at cubemap face boundaries? Is there a conflict between λc (global scale alignment) and local smoothness constraints (Eq 6)?
3. Provide a robustness table for (η_p, η_d, η_n, σ_int, σ_spa) across datasets, or propose an automatic heuristic.

**Ethical Concerns:**

["NO or VERY MINOR ethics concerns only"]

**Limitations:**

yes—Authors honestly mention smoothing of thin structures and slower inference vs feed-forward networks.

**Paper Formatting Concerns:**

The submission follows the NeurIPS template; no over-length, missing references, or figure issues detected.

**Quality:**

3

**Strengths And Weaknesses:**

The method is solid and yields large 360-depth gains without retraining, giving high practical value; yet novelty is mostly λ_c + graph optimisation, with no runtime/memory/hyper-parameter study, limited performance on thin structures, no comparison to recent diffusion approaches, and some notational clutter.

---

> ### Author Rebuttal · Authors · 2025-07-29
>
> We sincerely appreciate your thoughtful and constructive feedback.
> We are glad that our method was found to be solid and effective for 360 depth estimation without retraining, and we appreciate the recognition of our graph optimization with per-face scale alignment as the core contribution.
>
> ## **Q1. Time and Memory Usage**
> The **Table** showing inference time with respect to iteration count is included in our response to **Reviewer s1NL**’s **Q1**.
> However, allocating more iterations to high-resolution scales does not always yield better performance, as seen in the **fine** configuration.
> Our default setting (**original**), which follows a coarse-to-fine strategy (more iterations at lower scales, then refinement at higher scales), achieves the best balance between performance and efficiency.
> Depending on the application, one may reduce the total number of iterations (**half**) for faster inference with reasonable performance, or slightly improve performance by increasing iterations (**double**), though performance gains begin to saturate.
>
> Regarding memory usage, our method requires approximately 1680 MiB on an RTX A5000 GPU with an 8-core CPU when processing a $1024 \times 512$ panorama.
>
> ## **Q2. The role of $\lambda_c$**
> The overall optimization process can be interpreted as performing affine transformation of the form
> $$D^* = \lambda D + \tau ,$$
> where $D$ is the initial input depth and $D^*$ is the final consistent depth (**Line 141**).
> In this formulation, the per-face scale parameter $\lambda_c$ in Eq. 8 serves as the global scaling factor $\lambda$, which aligns the depth across cubemap faces.
> In contrast, the local smoothness constraint Eq. 6 effectively corrects the pixel-wise offsets, acting to estimate the bias term $\tau$.
>
> As a result, rather than conflicting, these two components work in a complementary fashion.
> $\lambda_c$ enforces global consistency across faces, while the local constraints refine local geometric coherence.
> Furthermore, Eq. 6 serves as the main loss function, whereas Eq. 8 acts as a regularization term that prevents the optimization from diverging too far from the input or collapsing to trivial solutions.
>
> ## **Q3. Hyperparameters**
> Following Rossi et al. [50], we set $\sigma_{int} = 0.07$ and $\sigma_{spa} = 3.0$.
>
> We conducted a grid search to empirically determine the hyperparameters $\eta_p = 50$, $\eta_d = 0.5$, and $\eta_n = 10$.
> The table below shows the performance with different hyperparameter values. While there are some variations, the overall performance distribution remains relatively stable.
>
> | Type     | ($\eta_p$, $\eta_d$, $\eta_n$) | Chamfer ↓ | a1 ↑ |
> |:----------|:------------------:|:----------:|:----------:|
> | original | (50, 0.5, 10)    | 0.33827       | 0.85503      |
> | | | | |
> | $\eta_p$ ↓       | (10, 0.5, 10)    | 0.35356  | 0.85073  |
> | $\eta_p$ ↑      | (100, 0.5, 10)   | 0.33572  | 0.85157  |
> | $\eta_d$ ↓      | (50, 0.05, 10)   | 0.36480  | 0.82762  |
> | $\eta_d$ ↑      | (50, 5.0, 10)    | 0.35525  | 0.84832  |
> | $\eta_n$ ↓      | (50, 0.5, 5.0)   | 0.33533  | 0.85569  |
> | $\eta_n$ ↑      | (50, 0.5, 100)   | 0.34166  | 0.85307  |
>
> Here, $\eta_p$ controls the weight of the local planar approximation, while $\eta_d$ and $\eta_n$ constrain the optimized depth from deviating too far from the original input.
> Among these, we observed that the optimization process is generally robust to $\eta_p$ and $\eta_n$, but slightly more sensitive to $\eta_d$.
>
> If $\eta_d$ is set too high, the optimization resists deviation from the original input, resulting in almost no difference between the input depth and the optimized depth.
> Consequently, the discrepancies between cubemap faces persist.
>
> Conversely, if $\eta_d$ is too small, the influence of the per-face parameters diminishes, weakening the constraint from the original input. This causes over-smoothing across cubemap faces and structural collapse near seams (as illustrated in **Fig. 6**, "**without $\lambda$**").
>
> Ultimately, we selected the hyperparameter values reported in the main paper because we believe that qualitative results are as important as numerical performance in 3D vision tasks, and the chosen values produced strong visual results.
> To ensure fair and consistent evaluation, we used the same set of parameters across all datasets and found that these values led to stable performance.
> While we cannot include figures in the rebuttal, we plan to add additional visualizations in the supplementary material, similar to **Figure 6**, to illustrate the effects of different hyperparameter settings.

---

> ### Comment · Reviewer_78uu · 2025-08-05
>
> Thank you for your detailed rebuttal. I still have one more question and hope to get your answer.
>
> As far as I know, some current depth estimation models will produce relatively large distortions in the depth estimation of the upper and lower surfaces (i.e., the ceiling and the floor) when dealing with panoramic depth estimation. This is specifically manifested as the depth of the ground or the sky being curved. Does the method you proposed have any optimizations in this regard, or can it solve some of the problems in this area? I haven't used Metric3Dv2, but I have shown distortion in this regard when using Depth Anything v2 and Omnidata v2. And could you please explain this aspect?

---

> > ### Author Response · Authors · 2025-08-05
> >
> > Thank you for your thoughtful question. Addressing the distortion issues (i.e., the ceiling and floor) in panoramic depth estimation is indeed one of the key challenges our work aims to solve.
> >
> > Perspective-based foundation models, such as Depth Anything v2, Omnidata v2, and Metric3D v2, are trained on undistorted images. As a result, when applied to distorted 360 equirectangular images, these models often produce depth inaccuracies in the curved regions.
> >
> > To address this, our approach first converts the input from an equirectangular projection (which contains distortion) into a **distortion-free cubemap** format. We then apply a foundation model to estimate the depth for each cubemap face. However, because monocular models predict each face independently, the resulting depth maps often lack inter-face scale consistency. To solve this, our main contribution introduces **per-face scale parameters** and a **graph optimization** technique that enforces **scale consistency across faces** in 3D space. This pipeline not only mitigates distortion but also induces scale consistency across views, enabling accurate 360 depth estimation.
> >
> > In conclusion, by utilizing image transformation for distortion removal, our method is compatible with various depth estimation foundation models, including Omnidata v2 and Metric3D v2 (as shown in **Table 3**), and demonstrates robustness to distortion in 3D space (as shown in **Figures 4 and 6**).

---

> > > ### Comment · Reviewer_78uu · 2025-08-05
> > >
> > > Thank you for your answer. I have understood what you mean. In fact, what I want to know is whether your method has a reliable optimization effect when the depth output of the base model is **already curved and discontinuous**. The existing perspective depth estimation models rarely have depth training data **directly facing the ground or directly facing the sky**. This often leads to depth curvature in perspective depth estimation when dealing with depth estimation of images directly facing the ground or the sky. However, for the six cubemaps of the panoramic view, there will definitely be one perspective view facing the ground and one perspective view facing the sky. Ominidata v2 uses **multiple views containing the same point** to estimate the depth of this point, eliminating this distortion to a certain extent. Even so, the Ominidata v2 still shows **obvious ground bending** in many cases. I would like to know how your method ensures the consistency between the depth of images directly facing the ground or sky, which are inherently inaccurate in depth estimation, and the gt depth when **only the base model is used for depth prediction once**?

---

> > > > ### Author Response · Authors · 2025-08-05
> > > >
> > > > As the reviewer mentioned, depth estimations from models like Omnidata v2 and Metric3D v2 can be inaccurate when the images are directly facing the ground or the sky. While I am unsure of the exact dataset the reviewer is using, we have experimented with datasets such as Matterport3D, Stanford2D3D, 360Loc, as well as a custom dataset.
> > > > In our experiments, normal estimation (to find the surface orientation per pixel) is relatively easier than depth regression and works better even for scenes directly facing a plane.
> > > > Both Omnidata v2 and Metric3D v2 offer normal estimation networks. This means that even when depth information is distorted (e.g., bending of the ground), normal information can be used to compensate for the distortion through joint graph optimization.
> > > >
> > > > Our optimization framework also addresses the bending problem directly.
> > > > We define the edge information for graph optimization using only pixel position and pixel value (**Eq. 7**), without relying on depth or normal.
> > > > This helps avoid issues when depth or normal estimations are inaccurate or bent.
> > > > With this definition, the graph optimization (**Eq. 6**) applies local planar approximation.
> > > > Here, by constraining neighboring normal values (those that are close in pixel position and have similar values with high $w_{ij}$) to be consistent, we prevent distortion or bending in the depth estimation.
> > > >
> > > > Additionally, as seen in the **Process Visualization on the project page** referenced in the abstract of our paper, the 3D points of the ceiling or ground, which initially appeared bent or inaccurate, are corrected and flattened through the optimization process.

---

> > > > > ### Comment · Reviewer_78uu · 2025-08-06
> > > > >
> > > > > Thank you for your response. I choose to keep the rate.

---

### Official Review · Reviewer_s1NL · 2025-07-03

**Clarity:** 3
**Significance:** 3
**Originality:** 3
**Rating:** 5
**Confidence:** 4

**Summary:**

This paper proposes a depth map estimation method from omnidirectional(360) images. The method utilizes the strength of recent foundation models to estimate accurate depth maps. Since those foundation models are trained based on perspective images rather than omnidirectional images, the method starts with partitioning the image into cube maps. After that, the method estimates the ERP space scale map using graph optimization with depth maps and normal maps. By using scale, the final resulting depth map is obtained. The method is evaluated on three datasets, using standard 3D and 2D metrics.

**Questions:**

There is some missing information and details:
- What is the running time? How long does it take for inference? How long is the optimization time?
- Neither the main paper nor the supplementary material includes the estimated ERP space scale map. It may be correct, since the downstream tasks and depth map metrics, but providing this along with the initial/final results (Figure 2 in supp.) may help readers to understand what the updates are.
- When merging cube maps to the ERP image again, how is the interpolation handled? There may be possibilities for choosing an improper sampling method for such remapping, which can lead to artifacts in edge cases.

**Ethical Concerns:**

["NO or VERY MINOR ethics concerns only"]

**Final Justification:**

The rebuttal properly answered my questions. I'm positive about the paper when focusing on the method that attempts to utilize foundation models for 360 images. Thus, I keep my ratings. However, I also agree with the other reviewer's opinion on the concern for the comparison methods. As a result, I cannot strongly claim the acceptance of the paper, and there is room to lower the score to Borderline Accept.

**Limitations:**

yes

**Paper Formatting Concerns:**

No formatting issues.

**Quality:**

3

**Strengths And Weaknesses:**

The proposed method uses perspective foundation models for the omnidirectional image, which seems to work. The core contribution to this is graph optimization for scale alignment. The proposed graph formulation, weight estimation, and optimization equation seem technically sound. The paper also includes comprehensive comparisons and ablation experiments, including variations on foundation models (Table 5) and other partitioning methods (Table 3). The resulting quantitative metrics and images show the state-of-the-art performance. There are no major technical flaws; only minor considerations are listed in the Questions section below.

---

> ### Author Rebuttal · Authors · 2025-07-29
>
> We thank the reviewer for their time and careful review. We appreciate the recognition of the technical soundness of our graph optimization for scale alignment, as well as the thoroughness of our evaluations across models (Table 5), partitioning strategies (Table 3), and datasets.
>
> ## **Q1. Running time**
> The inference time of our method depends on the input image resolution and the number of optimization iterations.
> For example, with an image size of $1024 \times 512$ and multi-scale iterations (Line 209)  set to $(300, 150, 30)$, the total inference time is approximately 4.6 seconds.
> We summarize the relationship between the number of iterations and performance in the Table below.
>
> The total number of iterations is $300 + 150 + 30 = 480$. To analyze the impact of iteration distribution, we define several configurations with the same total iterations: allocating more iterations to lower resolutions (coarse), allocating more to higher resolutions (fine), and distributing iterations evenly across all scales (equal).
> We also include configurations with double and half the number of iterations for comparison.
>
> |  Type  |   # Iterations  | Chamfer ↓ | a1 ↑ | time (sec) |
> |:--|:--:|:--:|:--:|:--:|
> | original | (300, 150, 30)   | 0.33827  | 0.85503 | 4.5898 |
> | coarse   | (400, 40, 40)    | 0.33684  | 0.85288 | 4.4416 |
> | fine     | (40, 40, 400)   | 0.37439  | 0.82867 | 8.7129 |
> | equal    | (160, 160, 160)  | 0.34565  | 0.85124 | 6.1151 |
> ||||||
> | half     | (150, 75, 15)    | 0.34897  | 0.84841 | 2.5647 |
> | double   | (600, 300, 60)   | 0.33499  | 0.85550 | 8.5421 |
>
> We tested RPG360 with Metric3D v2 on the Matterport3D dataset.
> Due to rebuttal time constraints, we sampled 100 images from Matterport3D to show performance trends.
> We will include full evaluation metrics in the supplementary material of the final version.
>
> ## **Q2. Visualization of Scale Map**
> We originally believed the ablation in Table 5 and Figure 6 sufficiently demonstrated the scale map’s impact since it was simply used as an intermediate representation of scale parameters for optimization.
> We will add the estimated scale map to help readers better understand the effect of the global scaling.
>
>
> ## **Q3. Concerns about Interpolation**
> When merging the cubemap predictions back into the ERP space, we use nearest-neighbor interpolation for both depth and normal maps, which is a standard practice.
> While this method may not fully address interpolation deviations caused by projection distortions, the subsequent joint graph optimization helps compensate for such issues.
> In our experiments, this interpolation strategy did not result in noticeable performance degradation.

---

> > ### Comment · Reviewer_s1NL · 2025-08-06
> >
> > I appreciate authors for providing additional information. Running time information is valuable to include in the paper for usability. The answers to other concerns also make sense. I expect the authors to add that information to the paper if accepted.

---

> > > ### Author Response · Authors · 2025-08-08
> > >
> > > We will add the running time analysis and the scale map visualization results to the supplementary material in the final version. We sincerely appreciate the reviewer’s positive feedback and are glad that our responses have addressed the concerns.

---

### Decision · Program_Chairs · 2025-09-17

**Decision:**

Accept (poster)

**Comment:**

The paper presents a training-free method for 360-degree depth estimation by combining perspective foundation models with graph-based scale alignment. Strengths include technical soundness, robust zero-shot generalization across datasets, and practical benefits in downstream tasks. Concerns were raised about overstated originality, the fairness of comparisons with baselines trained on smaller datasets, and limited evaluation on outdoor or degraded imagery. The rebuttal addressed runtime and robustness questions convincingly, and the authors committed to clarifying model details and expanding evaluations. Overall, the method is well-motivated, impactful, and reproducible, leading to the recommendation of acceptance.